# Effect of Sleeper-Ballast Particle Contact on Lateral Resistance of Concrete Sleepers in Ballasted Railway Tracks

**DOI:** 10.3390/ma15217508

**Published:** 2022-10-26

**Authors:** Jafar Chalabii, Majid Movahedi Rad, Ebrahim Hadizadeh Raisi, Reza Esfandiari Mehni

**Affiliations:** 1Department of Structural and Geotechnical Engineering, Széchenyi István University, Egyetem tér 1, 9026 Győr, Hungary; 2Islamic Azad University, Amol 678, Mazandaran, Iran; 3Department of Railway Track & Structures Engineering, School of Railway Engineering, Iran University of Science and Technology, Narmak, Tehran 13114-16846, Iran

**Keywords:** discreet element modelling, sleeper shape, lateral resistance, ballasted tracks, buckling

## Abstract

Although a sleeper makes a great contribution to the lateral resistance of ballasted tracks, in this regard, limited studies have been carried out on the effect of its contact surface with ballast aggregates. The current paper is dedicated to evaluating the effect of sleeper shape on the lateral resistance of ballasted track through discrete element modelling (DEM). For this purpose, firstly, a DEM model was validated based on the experimental results. Then, a sensitivity analysis was undertaken on the effect of the different contact areas that a standard concrete sleeper faces with the crib, shoulder and underlying ballast aggregates on lateral resistance of a single sleeper. As the main result of the current study, a high accurate regression equation for constant weight 319.2 kg and constant density 2500 kg/m^3^ of the sleepers was fitted between different sleeper contact areas and the maximum lateral resistance of a concrete sleeper for 3.5 mm lateral displacement in ballasted railway tracks. The obtained results showed that the effect of the sleeper’s head area compared to the underlying area of the sleeper and the head area of the sleeper compared to the sleeper’s side area in terms of lateral resistance are 8.2 times and 14.5 times more, respectively.

## 1. Introduction

Most of the railway lines worldwide are ballasted tracks thanks to easy maintenance and primary investments in this type of tracks. In general, this type of track consists of rails, fastening systems, sleepers, ballast and sub-ballast layers, and subgrade. Sleepers are a key component in the ballasted tracks as they are responsible to distribute the exerted loads from the rails to the underlying ballast layer. Sleepers also make a great contribution to the lateral resistance of the tracks depending on their materials, dimensions and interaction with ballast aggregates.

Nowadays, the construction of continuously welded rails (CWR), with the purpose of eliminating rail joints and, consequently, smoother operation and passengers’ comfort, has been developing. However, lateral resistance of CWR tracks due to buckling phenomena is still a challenge for engineers and researchers. Thus far, a growing body of railway track research has been allocated to this issue, especially the effect of ballast materials and its relevant topics on lateral resistance of ballasted tracks [1,2,3,4,5,6]. In the following, two of the cited papers, which are correlated with the ballast mechanism in lateral resistance, are explained in detail. 

Zhang et al. carried out research including field and three-dimensional DEM studies on lateral resistance of ballasted tracks affected by windblown sand. The results of in situ measuring indicated that windblown sand tracks bed shows an increase of 40% in terms of lateral resistance compared to clean ballasted tracks. It was also shown that the thicker the sand intrusion layer, the higher the lateral resistance and sharing ratio of ballast shoulder is [1]. With the purpose of increasing the lateral resistance of CWR tracks, the effect of geogrid reinforcement on lateral resistance of ballasted railway tracks was assessed using a study including single tie push tests (STPTs) and track panel displacement tests (TPDTs). From STPTs, lateral resistance increases of 31% and 42%, compared to unreinforced ballast, was reported for geocell-reinforced ballast by one and two geogrid layers. In the same way, these magnitudes were 29% and 49% for TPDTs [7]. 

Some technical papers, on the other hand, are dedicated to assessing the contribution of the sleeper and its associated solutions to the lateral resistance of ballasted tracks [8,9,10]. For instance, Zakeri et al. [11] investigated the effect of three sleeper shapes of conventional concrete B70, winged and friction on mitigation of buckling in CWR tracks using both filed and numerical tests. The results indicated that the lateral resistance of winged and frictional sleepers were higher than conventional B70 sleeper by 49% and 68%, respectively. It was also revealed that for a 100 m radius tight curve, using frictional concrete sleepers was the only alternative to resist track buckling. In another innovative study conducted by Esmaeili et al. [12], the effect of concrete nailed sleepers was examined using a developed 3D finite element model. In fact, in this method, the lateral resistance potential of the subgrade is engaged using nails along the sleepers. The outcomes revealed that the length of nails was effective on lateral resistance of a nailed sleeper up to 1 m and remains constant for longer nails. Nail diameters of 16, 24, 32 and 40 mm had resulted in increasing lateral resistance by 55, 90, 120 and 150%, respectively, in comparison with conventional B70 sleepers. The distance of 40 cm from the sleeper center is also recognized as the optimum location for nails in terms of increasing lateral resistance. 

Although different numerical methods have been widely used for modelling ballasted railway tracks, DEM has received special attention among them for considering mechanical interactions between ballast aggregates themselves and with sleepers [13,14,15,16]. The effect of sleeper bottom face texture on lateral resistance was examined by Guo et al. [17] using DEM simulation. For this purpose, four concrete sleeper types, including three innovative frictional and a mono-block sleeper, were studied to determine the optimal type in terms of lateral resistance. It was concluded that frictional sleepers have better performance as more aggregates are contributed to the lateral resistance of the track. In the same study [18], the lateral resistance of concrete sleepers with an arrowhead groove was studied adopting both STPT and DEM analysis. An interesting result of this study was that arrowhead groove sleepers with a ballast shoulder of 30 cm can provide approximately the lateral resistance as the same normal sleeper with a 50 cm ballast shoulder which can be used in special super-structures where there is a space limitation. Although steel sleepers have not been widely used in ballasted tracks, in 2018, the lateral displacement of different steel sleeper types placed in a ballast layer with different shoulder heights and widths was investigated through both STPTs and numerical modelling [19]. DEM results showed that in the case of using steel sleepers, the ballast bed had a higher contribution to the lateral resistance compared to the other ballast components. Recently, a DEM analysis was carried out to examine the lateral resistance of different concrete sleepers under various types of support [20]. Based on DEM results, using HA110, winged and middle-winged sleepers resulted in an increase of 38, 45 and 22% in terms of lateral resistance compared to conventional B70 sleepers.

Zhu et al. [21] evaluated the stability of the asphalt mixture aggregate network and the mechanism of coarse aggregate movement of the asphalt mixture during the compaction process. The result showed that the particles sizes affect the unbalanced force distribution of the coarse aggregate, and with increasing blows, the coarse aggregate’s imbalance force reduced, indicating that compaction has a tendency to make the asphalt mixture more stable. The macroscopic and microscopic behavior of the GBM was evaluated by Li et al. under repeated load triaxial testing with the discrete-element method [22]. The findings demonstrated good agreement between the resilient modulus determined by the developed simulation and that determined by laboratory measurements. The resilient modulus of the GBM increases from 94.03 to 337.71 MPa with the growth of the volume stress and deviator stress, which is a stress-hardening feature of the GBM’s deformation behavior. More stresses are distributed throughout the skeleton as a result of the strong mechanism it offers for resistance to deformation. In 2020, Lim et al. [23] proposed a rational method for track ballast–wheel interaction that may well be advanced and created to demonstrate the interaction in a train-derailment occasion, based on the discrete-element strategy. The recommended analysis method was approved through comparison with the experimental results of a drop test. In addition, case studies were conducted to examine the impacts of the contact-model parameters on the simulation result.

Having a comprehensive review of technical literature, it was clarified that in spite of the importance of the interaction surface area of concrete-sleeper contact with ballast particles, limited studies have been carried out in this regard so far. Therefore, the current study’s aim is to properly respond to the question of what relationship is between the surface area of sleeper-ballast contact and lateral resistance of standard concrete sleeper, B70. Hence, in the first step, a DEM model was validated in the PFC 3D [24] environment using the experimental data of STPTs carried out previously [16]. It should be mentioned that, in this study, the DEM method for simulation has been selected since in this method the behavior of granular materials can be considered. This is despite the fact that other methods such as simulation in Abaqus do not have the possibility to check granular behavior.

Afterwards, a sensitivity analysis was conducted on concrete sleeper dimensions, where they are in contact with ballast aggregates, to determine the correlation of maximum lateral resistance with the lateral displacement of 3.5 mm and different contact faces of the sleeper with ballast particles. In other words, a relationship is obtained between the different contact faces of the sleeper and the maximum lateral force by keeping weight, density and maximum displacement value of the sleeper constant.

## 2. Material Specifications and DEM Simulations

In current section, the steps of DEM modelling are described as well as specifications of different elements used in the model. In the following, firstly the sleeper and ballast aggregates properties are addressed. Afterwards, the contact model is described as an important component in DEM simulations. This section finally ends with SPTP simulation and verification using previously provided data in a study carried out by [19].

### 2.1. Ballast

The ballast material gradation used in this study was chosen in accordance with AREMA No. 24 and given particle size distribution in Figure 1. Similar to the experimental material, the density and porosity of ballast layer were also 1700 kg/m^3^ and 0.35, respectively. 

One of the DEM modeling methods for ballast particles is to generate spheres [25]. Application of a sphere reduces the computation time in addition to simplified calculations. However, due to the low rolling resistance of the spheres, more deformation is expected than the real ballast particles which result in low shear strength. The RR-Linear (Rolling Resistance Linear) contact model can be used to solve this problem [26]. Therefore, in this study, simple spherical balls were used in discrete element models to simulate ballast aggregates.

### 2.2. Sleeper

Concrete sleeper type B70, which has been commonly using in ballasted railway tracks, was selected for numerical simulations. Figure 2 shows the dimensions of the sleeper and the simulated sleeper in PFC 3D. It should be noted that to help make a proper shape of the sleeper, it was firstly used as a STL file and then added PFC3D as input. The clump elements were used to generate the sleeper because of governing the clump logic of spherical aggregates. Figure 2b illustrates the generated sleeper consisting of 2888 pebbles. Using the clump element instead of wall boundaries for generating the sleeper gives the advantage of force assignment [16]. The maximum inner pebble angular measure smoothness was adopted to minimize sleeper surface roughness as was discussed in detail in a study carried out by Taghavi, 2011 [27]. Consequently, in the current study, $\propto =180^{{}^{\circ}}$, which characterizes a quite smooth contact, was chosen for generating the sleeper using clumps.

### 2.3. Contact Model

A model of linear elastic contact is considered to simulate and calculate the contact forces generated between wall-particle contacts and inter participle connections. Figure 3 depicts a schematic of this contact in accordance with guideline published by PFC 3D, Itasca [24]. Details of the linear elastic contact is addressed in Table 1. It should be noted that this contact model was used for all types of connections including ball–Pebble (ballast–sleeper), ball–ball (ballast–ballast) and ball–Facet (ballast–wall).

According to Figure 3a, $F_{n}^{l}$ and $F_{s}^{l}$ indicate the normal and shear components of the linear force, respectively. Additionally, $F_{n}^{d}$ and $F_{s}^{d}$ signify the normal and shear components of the dashpot force, separately. In addition, *$\beta_{n}$*, $\beta_{s}$, $g_{s}$, and μ show normal critical damping ratio, shear critical damping ratio and friction coefficient, respectively [24]. 

### 2.4. DEM Simulation

Concrete sleeper type B70, which has been commonly using in ballasted railway tracks, was selected for numerical simulations. Figure 2 shows the dimensions of the sleeper and the simulated sleeper in PFC 3D. It should be noted that to help make a proper shape of the sleeper, it was firstly generated as an STL file and then called in PFC3D as input. The clump elements were used to generate the sleeper because of governing the clump logic of spherical aggregates. Figure 2b illustrates the generated sleeper consisting of 2888 pebbles. Using clump elements instead of wall boundaries for generating the sleeper gives the advantage of force assignment [16]. 

The maximum inner pebble angular measure smoothness was adopted to minimize sleeper surface roughness, as was discussed in detail in a study carried out by Taghavi, 2011 [27]. Consequently, in the current study, $\propto =180^{{}^{\circ}}$, which characterizes a quite smooth contact, was chosen for generating the sleeper using clumps. The running time of the model shown in Figure 4 took approximately 48 h. The run model is shown in Figure 5.

## 3. Results and Discussion

### 3.1. Validation

To develop a 3D DEM model, the first step was to validate the results of simulated STPT with those obtained in the filed experiments carried out by Khatibi et al. [16]. It should be mentioned that in their study, three different STPTs were conducted on a common ballasted test track located in southwest of Tehran. Figure 6 shows DEM simulation results versus the experimental results. In this figure, the load-displacement response resulting from DEM simulation is compared to the field results linked to a ballasted railway track with ballast depth, shoulder width, friction coefficient and porosity of 35 cm, 40 cm, 0.9 and 0.35, respectively. From the figure, good agreement can be seen between the curves with the maximum difference of 7.9% for lateral force related to lateral displacement of 3.5 mm. This difference can be linked to the dissimilarities of ballast and sleeper shapes and test condition.

In addition, the precision of simulated of lateral resistance was further investigated graphically, which is illustrated in Figure 7. To better understand, the more the results are close to 1:1 line, the more precision will have the equations. Considering the standard error reported by Shahin (2015) [28], a good visual judgment can be made through two dashed lines with ±5° deviation from the perfect agreement.

### 3.2. Parametric Study

Based on the validated STPT DEM model, comprehensive sensitivity analyses are carried out on the effect of the concrete B70 sleeper’s shape on lateral resistance of ballasted railway tracks. To this end, all parameters and specifications considered in validated STPT stayed constant and the only parameters that changed was the shape of the sleeper. Consequently, after applying the lateral force in accordance with the steps presented in validated model, lateral forces are recorded for a lateral displacement of 3.5 mm.

Figure 8 illustrates different faces of the sleeper which are in contact with ballast aggregates as well as effective shear resistances. From equilibrium of forces in z and x directions:(1)$$\sum F_{z}=0\;\overset{so}{\Rightarrow }\;W=N$$
(2)$$\sum F_{x}=ma\;\overset{a≅0}{\Rightarrow }\;\sum F_{x}=0\;\;\overset{so}{\Rightarrow }\;F-\sigma A_{f}-2\tau_{2}A_{a}-\tau_{1}A_{b}=0$$
(3)$$\overset{so}{\Rightarrow }\;F=\sigma A_{f}+2\tau_{2}A_{a}+\tau_{1}A_{b}$$
where σ depends on PSD material properties of ballast and sleeper. On the other hand, the following relationship is true:(4)$$\tau_{2}\propto \mu N\;\left(N=W\right)$$
where *μ* is friction compact and depends on material properties of ballast and sleeper. Therefore, $\tau_{2}$ depends on the sleeper’s weight and material properties of ballast and sleeper. As PSD, sleeper’s weight and material properties of ballast and sleeper are constant, it can be concluded that σ, $\tau_{2\;}$ and $\tau_{1}\;$ will remain constant. Thus:(5)$$F=Constant1.A_{f}+Constant2.A_{a}+Constant3.A_{b}$$

### 3.3. Regression Equations

Finding a relationship between friction in ballasted railway tracks and ballast-sleeper contacts has always been a challenge for railway and geotechnical engineers and researchers. Thus far, some studies have been dedicated to this topic [9,12,13].

Although many attempts have been made for solving this issue, there is no study specifically focused on a direct relationship between lateral resistance of concrete sleepers and ballast-sleeper contact area. In this study, in order to find a proper correlation between the lateral resistances of the sleepers, the data provided in Table 2 were statistically analyzed. Hence, several regression equations were generated for lateral resistance as a function of different faces of sleeper interacted with ballast aggregates, $A_{f}$ (head areas), $A_{b}$ (underlying contact area) and $A_{a}$ (side areas). The result shown in Table 2 is for lateral displacement 3.5 mm and the weight has been considered constant. For determining the best prediction equation, two factors were controlled, including low disagreement between equation and DEM data and $R^{2}$ value close to 1.0. To clarify, the more the data are close to 45-degree line, the more accurate is the predicted equation. From Figure 9, by considering standard error of ±5°, dash lines, it can be seen that majority of the spots are placed between this domain, which shows a good confirmation between driven equation and collected data from DEM analyses.
(6)$$F_{max}\left(\mathrm{KN}\right)=\left(67.33A_{f}\left({\mathrm{mm}}^{2}\right)+4.654A_{a}\left({\mathrm{mm}}^{2}\right)+8.1905A_{b}\left({\mathrm{mm}}^{2}\right)\right)\times 10^{-6}\;\;\;\;\;\;\;\;R^{2}=0.97$$

## 4. Conclusions

The current study was dedicated to investigate the effect of ballast-sleeper contact on lateral resistance of common concrete sleepers in ballasted railway tracks. To do this, firstly, a DEM model of an experimental STPT was modelled three-dimensionally in PFC 3D software version of 4. Then, the lateral-displacement–lateral-resistance response of the DEM model was validated using experimental data already provided by Khatibi et al. [16]. Afterwards, on the basis of this validated model, a series of sensitivity analysis was carried out on different faces of sleeper, which are in contact with ballast aggregates. The summary of the obtained results are as follows:The DEM results show a good agreement with experimental data in terms of lateral resistance displacement. However, there was a gap between the graph, which is naturally associated with dissimilarities of ballast and sleeper’s shapes and also the loading process and conditions;A regression equation was obtained to predict the lateral resistance of a concrete sleeper as a function of three different surface areas of sleeper interacted with ballast aggregates. The accuracy of this equation was about 92%, which is reliable enough to be used in this regard;According to Table 2, the changing trend of *F_max_* from sleeper B-0 d to sleeper B-5 is decreasing, the area value of *A_f_* is increasing and *A_a_*, *A_b_* is decreasing. It shows that from B-0 to B-5, the downward trend of the *A_a_
* and *A_b_* in the downward trend of fmax has more impact than *A_f_*. This is while the upward trend has started from B-6 and continues to B-16, which shows the greater influence of *A_f_* in this trend. In addition, equation 6 obtained from Table 2 shows the high influence of *A_f_* compared to *A_a_* and *A_b_* in *F_max_*. The effect value of *A_f_* in *F_max_* compared to *A_b_* and *A_a_* is 8.2 times and 14.5 times more, respectively;For recommending a sleeper, first of all, according to the standard gauge of the railway, which is 1433 mm [29], the suggested dimensions of the sleeper should be determined based on this gauge. As shown in Table 2, sleepers of B-0 to B-8 are 2889 mm to 1502 mm in length, respectively, and are suitable for supplying standard gauge. In addition, based on the results of maximum forces in Table 2, the sleeper of B-0 is recommended.

However, it should be noted that there were some limitations in this research, for example, the shape of ballast particles was assumed spherical. Although this problem was compensated by the sliding resistance to an acceptable extent according to the results, taking into account the real shape of the particles would have provided more accurate results. In addition, to simplify, the mass of the sleeper was assumed constant; in future studies, the effect of the weight of the sleeper on lateral resistance can be studied to provide an optimum sleeper’s weight by considering the standards.

## Figures and Tables

**Figure 1 materials-15-07508-f001:**
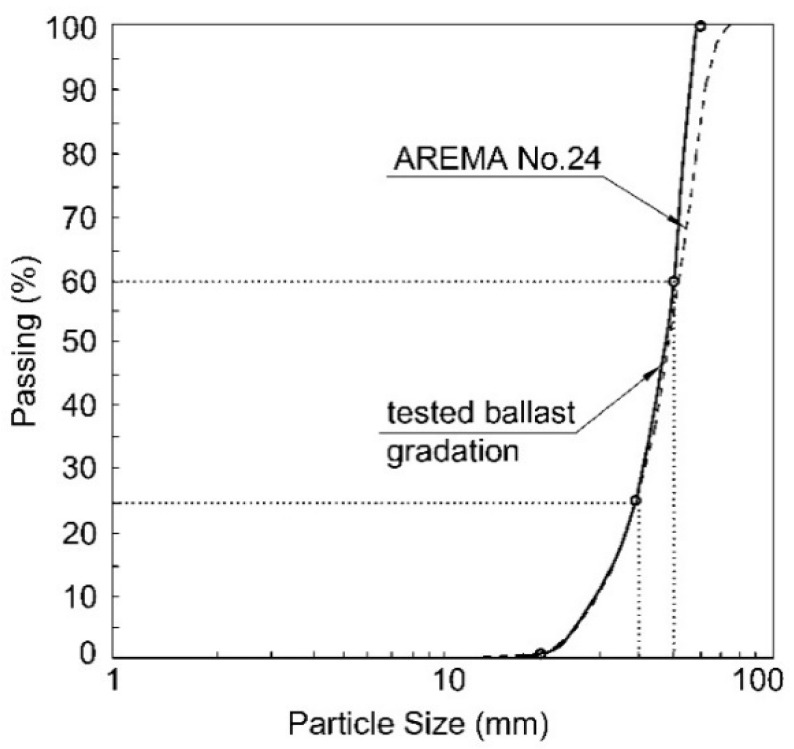
Ballast particle size distribution curve.

**Figure 2 materials-15-07508-f002:**
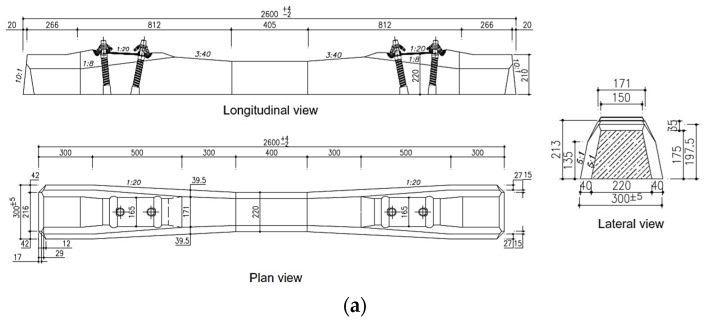
Used sleeper in DEM (**a**) dimensions in mm (**b**) simulated sleeper.

**Figure 3 materials-15-07508-f003:**
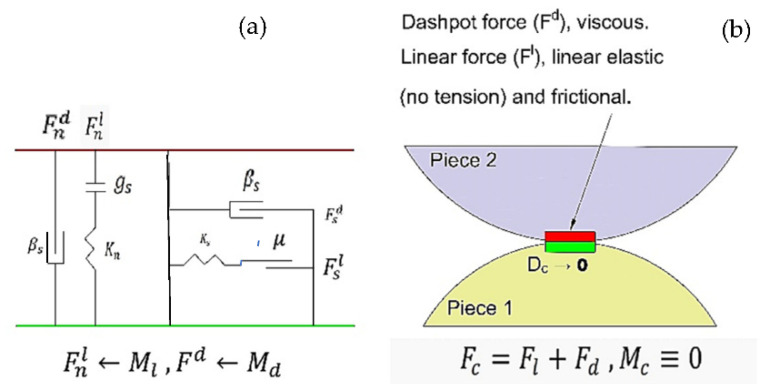
Behavior and rheological components of the linear model. (**a**) A schematic of vertical and horizontal forces of the linear model. (**b**) A schematic of linear contact of two particles [24].

**Figure 4 materials-15-07508-f004:**
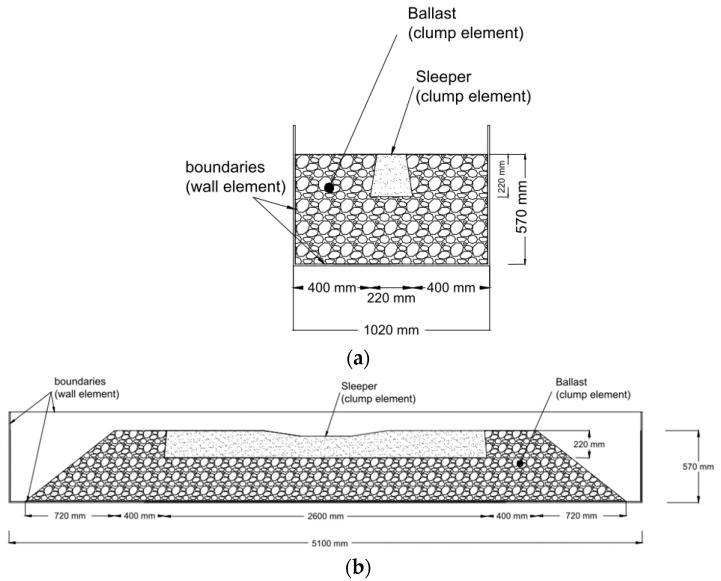
STPT schematic (**a**) cross section (**b**) longitudinal section.

**Figure 5 materials-15-07508-f005:**
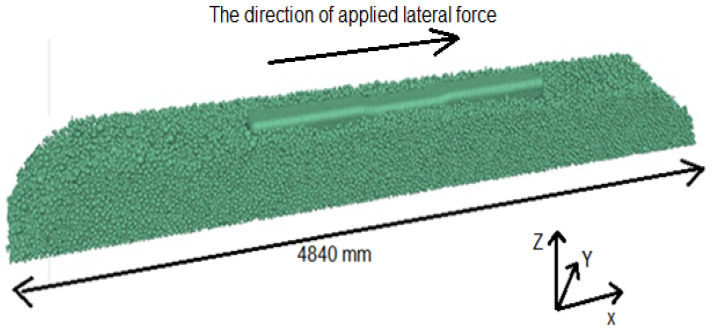
Simulated STPT model in PFC 3D.

**Figure 6 materials-15-07508-f006:**
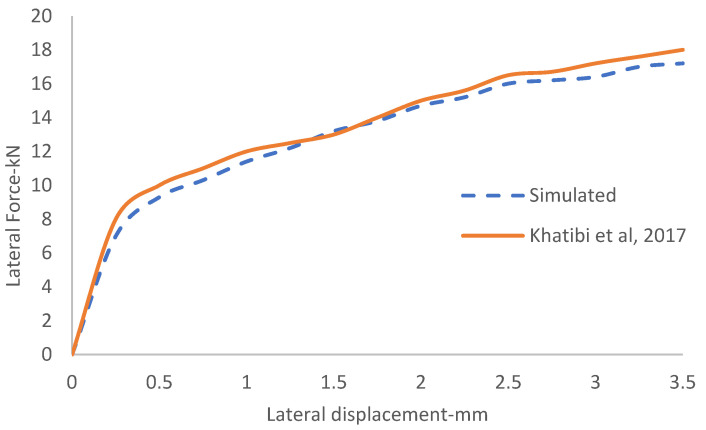
Comparison of the simulated values of lateral force vs. the laboratory values of lateral force [16].

**Figure 7 materials-15-07508-f007:**
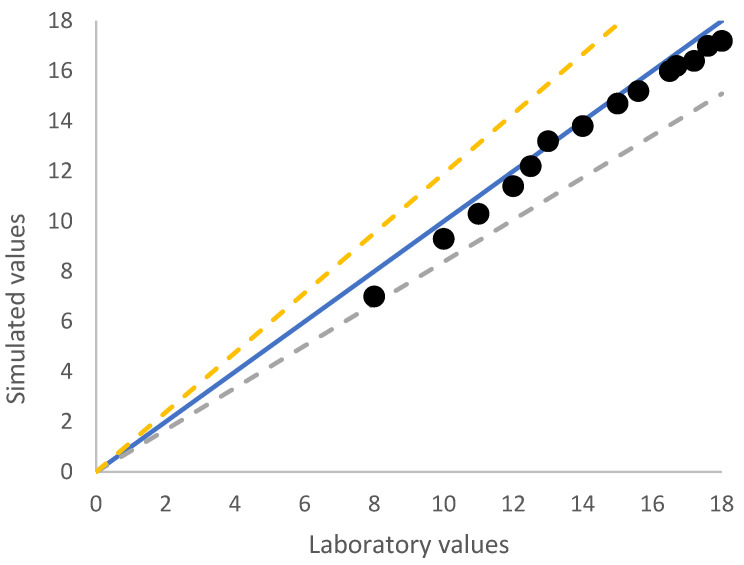
Lateral force–displacement curve of simulated compared to experimental data.

**Figure 8 materials-15-07508-f008:**
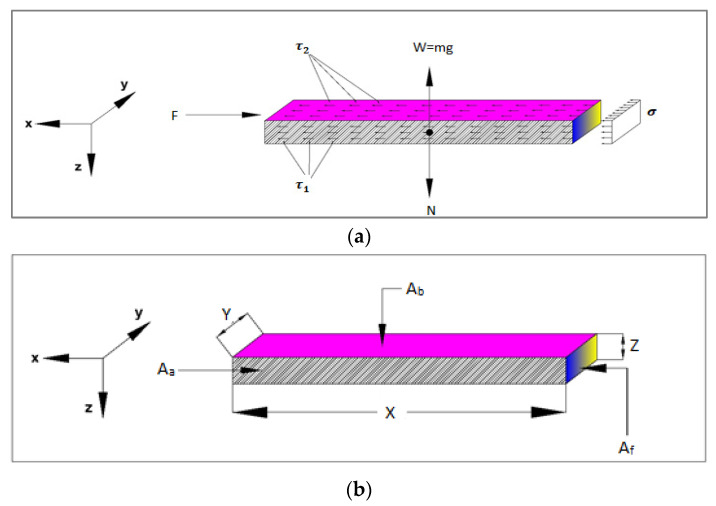
Schematics of (**a**) sleeper dimensions (**b**) governing forces.

**Figure 9 materials-15-07508-f009:**
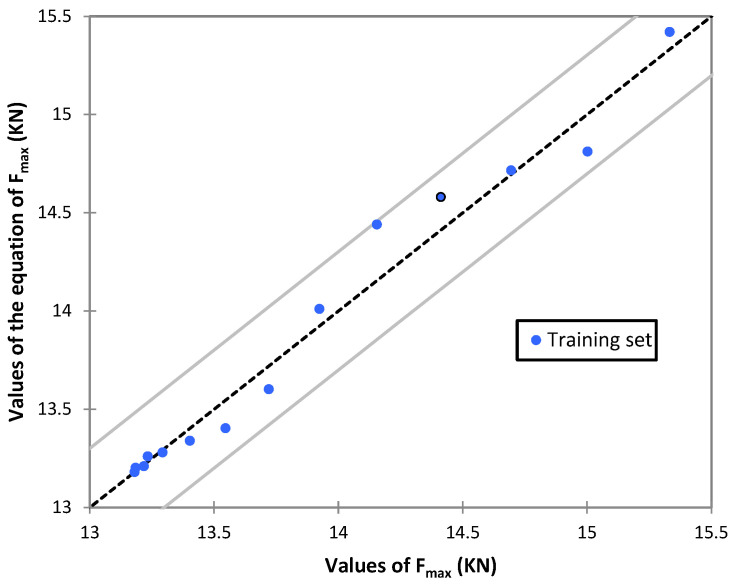
Comparison of the simulated values of maximum lateral force vs. the equation of maximum lateral force.

**Table 1 materials-15-07508-t001:** Contact parameters used in DEM simulation.

Particle and Contact Parameter	Symbol	Unit	Value
Ballast particle density	$\gamma_{b}$	$\mathrm{Kg}/m^{3}$	2600
Sleeper clump density	$\gamma_{s}$	$\mathrm{Kg}/m^{3}$	2500
Inter particle normal stiffness	$K_{n}$	Pa	$0.42\times 10^{8}$
Inter particle shear stiffness	$K_{s}$	Pa	$0.55\times 10^{8}$
Inter particle coefficient of friction	FI	--	0.9
Wall normal and shear stiffness	$K_{s},\;K_{n}$	Pa	$1\times 10^{8}$
Base wall friction coefficient	$f_{wb}$	--	0.57
Side wall friction coefficient	$f_{ws}$	--	0.9

**Table 2 materials-15-07508-t002:** Lateral forces obtained based for different sleeper dimensions.

Number	Sleeper	X (mm)	Y (mm)	Z (mm)	Volume (mm^3^)	A_f_ (mm^2^)	A_b_ (mm^2^)	A_a_ (mm^2^)	F_max_ (KN)
1	B-0	2889	260	170	127,680,000	44,200	751,059	982,154	13.600
2	B-1	2627	270	180	127,680,000	48,600	709,333	945,778	13.480
3	B-2	2400	280	190	127,680,000	53,200	672,000	912,000	13.404
4	B-3	2201	290	200	127,680,000	58,000	638,400	880,552	13.260
5	B-4	2027	300	210	127,680,000	63,000	608,000	851,200	13.202
6	B-5	1872	310	220	127,680,000	68,200	580,364	823,742	13.180
7	B-6	1735	320	230	127,680,000	73,600	555,130	798,000	13.210
8	B-7	1612	330	240	127,680,000	79,200	532,000	773,818	13.280
9	B-8	1502	340	250	127,680,000	85,000	510,720	751,059	13.340
10	B-9	1403	350	260	127,680,000	91,000	491,077	729,600	13.403
11	B-10	1314	360	270	127,680,000	97,200	472,889	709,333	13.602
12	B-11	1232	370	280	127,680,000	103,600	456,000	690,162	14.010
13	B-12	1159	380	290	127,680,000	110,200	440,276	672,000	14.440
14	B-13	1091	390	300	127,680,000	117,000	425,600	654,769	14.580
15	B-14	1030	400	310	127,680,000	124,000	411,871	638,400	14.716
16	B-15	973	410	320	127,680,000	131,200	399,000	622,829	14.812
17	B-16	921	420	330	127,680,000	138,600	386,909	608,000	15.421

## Data Availability

The datasets generated during and/or analyzed during the current study are available in the main manuscript.

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
