# Peer review of "Effect of Sleeper-Ballast Particle Contact on Lateral Resistance of Concrete Sleepers in Ballasted Railway Tracks"

_materials, 2022, doi:10.3390/ma15217508_

Round 1

Reviewer 1 Report

The manuscript entitled "Effect of sleeper-ballast particles contact on lateral resistance of concrete sleeper in ballasted railway tracks" provides an in-depth discussion of the effect of sleeper shape on lateral resistance of ballasted track through discrete element modelling (DEM). However, the manuscript has a few shortcomings, which should be address before considering publication.

1.      The literature review needs to be concise and clearly define how the paper address the gap in literature considering that lots of work conducted on discrete element modelling (DEM). Describe how the objectives are unique from the previous studies. Due to that, relevant study suggests to cited, as follow:

*Xuan Zhu, Guoping Qian, Huanan Yu, et al., Evaluation of coarse aggregate movement and contact unbalanced force during asphalt mixture compaction process based on discrete element method, Construction and Building Materials. 328 (2022) 127004. https://doi.org/10.1016/j.conbuildmat.2022.127004.

*Li J, Zhang J, Zhang A, et al. Evaluation on Deformation Behavior of Granular Base Material during Repeated Load Triaxial Testing by Discrete-Element Method [J]. International Journal of Geomechanics. 22(11) (2022) 04022210. https://doi.org/10.1061/(ASCE)GM.1943-5622.0002539.

*Lim N, Kim K, Bae H, et al. DEM Analysis of Track Ballast for Track Ballast–Wheel Interaction Simulation[J]. Applied Sciences. 10(8) (2020) 2717. https://doi.org/10.3390/app10082717.

2.      The contact parameters in Table 1 need to be checked, such as normal stiffness and shear stiffness should be 0.42e8. Figure 3 should be marked (a) (b) where symbols such as gs should indicate their physical meaning.

3.      Some additional information on computational effort, such as model size, computational time with respect to the employed machines, PC or workstation specifications as far as the numerical analysis are concerned, should at least be added

4.      Authors should provide some justification in choosing the Particle Flow Code for DEM simulation instead of other well-known commercial (such as EDEM or DEM tool in Abaqus system) and/or not commercial (such as YADE) codes already available.

5.      The proposed sleeper size should be given to improve the study availability. At the same time, the discrete element method is also a mesomechanical analysis method, and the influence mechanism of sleeper shape on lateral force should be analyzed by contact force, so as to provide a strong theoretical basis.

Author Response

Response to Reviewer 1:

The manuscript entitled "Effect of sleeper-ballast particles contact on lateral resistance of concrete sleeper in ballasted railway tracks" provides an in-depth discussion of the effect of sleeper shape on lateral resistance of ballasted track through discrete element modelling (DEM). However, the manuscript has a few shortcomings, which should be address before considering publication.

Response: Thank you very much.

  1. The literature review needs to be concise and clearly define how the paper address the gap in literature considering that lots of work conducted on discrete element modelling (DEM). Describe how the objectives are unique from the previous studies. Due to that, relevant study suggests to cited, as follow:

*Xuan Zhu, Guoping Qian, Huanan Yu, et al., Evaluation of coarse aggregate movement and contact unbalanced force during asphalt mixture compaction process based on discrete element method, Construction and Building Materials. 328 (2022) 127004. https://doi.org/10.1016/j.conbuildmat.2022.127004.

*Li J, Zhang J, Zhang A, et al. Evaluation on Deformation Behavior of Granular Base Material during Repeated Load Triaxial Testing by Discrete-Element Method [J]. International Journal of Geomechanics. 22(11) (2022) 04022210. https://doi.org/10.1061/(ASCE)GM.1943 5622.0002539.

*Lim N, Kim K, Bae H, et al. DEM Analysis of Track Ballast for Track Ballast–Wheel Interaction Simulation[J]. Applied Sciences. 10(8) (2020) 2717. https://doi.org/10.3390/app10082717.

Response: I appreciate that you suggested these sources. Considering these references was clarified that in spite of the importance of the interaction surface area of concrete-sleeper contact with ballast particles, limited studies have been carried out in this regard so far. It was also considered in the manuscript.

  1. The contact parameters in Table 1 need to be checked, such as normal stiffness and shear stiffness should be 0.42e8. Figure 3 should be marked (a) (b) where symbols such as gs should indicate their physical meaning.

Response: The mistakes in table 1 were solved and figure 3 was separated into (a), and (b). Also, the parameters in Figure 3 were described.

  1. Some additional information on computational effort, such as model size, computational time with respect to the employed machines, PC or workstation specifications as far as the numerical analysis are concerned, should at least be added

Response: Time of running was mentioned in the part of DEM simulation and also the size of model can be seen in figure 4.

  1. Authors should provide some justification in choosing the Particle Flow Code for DEM simulation instead of other well-known commercial (such as EDEM or DEM tool in Abaqus system) and/or not commercial (such as YADE) codes already available.

Response: That was mentioned in Introduction part.

  1. The proposed sleeper size should be given to improve the study availability. At the same time, the discrete element method is also a mesomechanical analysis method, and the influence mechanism of sleeper shape on lateral force should be analyzed by contact force, so as to provide a strong theoretical basis.

Response: I thank you for this incredible thought. That was proposed within the conclusion portion and a sleeper was proposed with specific size.

Reviewer 2 Report

Effect of sleeper-ballast particles contact on lateral resistance of concrete sleeper in ballasted railway tracks. Interesting article however, some comments mentioned bellow, 

1) The abstract is not clear. An abstract is a short summary of your completed research. It is intended to describe your work without going into great detail. Abstracts should be self-contained and concise, explaining your work as briefly and clearly as possible.

 2) Some more latest studies are required in the introduction section to further highlight the importance of this study.

3) The results and discussion are not clearly dealt the outcomes of the proposed work. The authors should explicitly state the novel contribution of this work, the similarities, and the differences of this work with the previous publications in this section.

4) It is suggested to highlight the limitations of this study, suggested improvements of this work and future directions in the conclusion section. Also, the conclusion can be presented better than the present form with more findings.

Author Response

Response to Reviewer 2:

Effect of sleeper-ballast particles contact on lateral resistance of concrete sleeper in ballasted railway tracks. Interesting article however, some comments mentioned bellow,

Response: Thank you exceptionally much.

1) The abstract is not clear. An abstract is a short summary of your completed research. It is intended to describe your work without going into great detail. Abstracts should be self-contained and concise, explaining your work as briefly and clearly as possible.

Response: Thank you for your comment, it has been corrected as much as possible, and tried to provide clear and concise information about our research.

2) Some latest studies are required in the introduction section to further highlight the importance of this study.

Response: Three references were included to advance highlight the significance of this study. The added references as follows:

*Xuan Zhu, Guoping Qian, Huanan Yu, et al., Evaluation of coarse aggregate movement and contact unbalanced force during asphalt mixture compaction process based on discrete element method, Construction and Building Materials. 328 (2022) 127004. https://doi.org/10.1016/j.conbuildmat.2022.127004.

*Li J, Zhang J, Zhang A, et al. Evaluation on Deformation Behavior of Granular Base Material during Repeated Load Triaxial Testing by Discrete-Element Method [J]. International Journal of Geomechanics. 22(11) (2022) 04022210. https://doi.org/10.1061/(ASCE)GM.1943 5622.0002539.

*Lim N, Kim K, Bae H, et al. DEM Analysis of Track Ballast for Track Ballast–Wheel Interaction Simulation[J]. Applied Sciences. 10(8) (2020) 2717. https://doi.org/10.3390/app10082717.

3) The results and discussion are not clearly dealt the outcomes of the proposed work. The authors should explicitly state the novel contribution of this work, the similarities, and the differences of this work with the previous publications in this section.

Response: That was mentioned in conclusion part.

4) It is suggested to highlight the limitations of this study, suggested improvements of this work and future directions in the conclusion section. Also, the conclusion can be presented better than the present form with more findings.

Response: The limitation of this study was proposed within the conclusion conjointly recommended a few recommendations for future think about.